# Prognostic Significance of the N-Terminal Pro-B-Type Natriuretic Peptide in Lung Transplant Candidates on the Waiting List

**DOI:** 10.3390/diagnostics12092112

**Published:** 2022-08-31

**Authors:** Shimon Izhakian, Assaf Frajman, Lev Freidkin, Osnat Shtraichman, Dror Rosengarten, Barak Pertzov, Yaron D. Barac, Mordechai Reuven Kramer

**Affiliations:** 1Pulmonary Institute, Beilinson Hospital, Rabin Medical Center, Petah Tikva 4941492, Israel; 2Sackler Faculty of Medicine, Tel Aviv University, Ramat Aviv 6997801, Israel; 3Division of Cardiovascular and Thoracic Surgery, Beilinson Hospital, Rabin Medical Center, Petah Tikva 4941492, Israel

**Keywords:** NT-proBNP, lung transplantation, pulmonary hypertension, prognosis, mortality

## Abstract

We investigated the prognostic significance of N-terminal pro-B-type natriuretic peptide (NT-proBNP) in lung transplant candidates, in a retrospective single-center study. Data regarding various baseline characteristics and all-cause mortality were collected for 205 lung transplant candidates placed on waitlist for transplantation from November 2017 to December 2019. Associations of NT-proBNP levels with baseline characteristics and mortality were analyzed. Results showed NT-proBNP values correlated positively with age, forced vital capacity, mean pulmonary artery pressure (MPAP), and pulmonary capillary wedge pressure; and negatively with diffusing lung capacity for carbon monoxide and cardiac index. The optimal cut-off of NT-proBNP for predicting MPAP levels > 35 mmHg was 251 pg/mL; with 58.1% sensitivity, 85.7% specificity, 45.0% positive predictive value, and 91.0% negative predictive value. During a median follow-up period of 2.2 years, 97 patients underwent lung transplantation, 42 died waiting for donation, and 66 were alive and still waiting for transplantations. On multivariate analysis, higher NT-proBNP levels were strongly associated with increased mortality among waitlisted lung transplant candidates (HR 1.49, 95% CI 1.10–2.03, *p* = 0.01). In conclusion NT-proBNP can predict mortality among waitlisted lung transplant candidates. Lower levels of NT-proBNP can preclude severe pulmonary artery hypertension. Assessment of NT-proBNP may improve risk stratification among lung transplant candidates.

## 1. Introduction

Brain natriuretic peptide (BNP) is mainly synthesized and secreted by myocytes in the cardiac ventricles, as a response to myocytes stretched by pressure overload or volume expansion of the ventricles [1]. Cleavage of the prohormone proBNP produces biologically active 32 amino acid BNP as well as biologically inert 76 amino acid N-terminal pro-BNP (NT-proBNP). NT-proBNP has a longer half-life, is considered more stable, and is less influenced by acute hemodynamic changes than BNP [1].

Heart failure (HF), cor pulmonale, secondary pulmonary hypertension (PH), and hypoxemia represent important stimuli for releasing natriuretic peptides from the heart and increasing BNP gene expression [1,2,3,4].

The prognostic significance of BNP was extensively studied in patients with class 1 PH (pulmonary arterial hypertension) [5]. In this setting, elevated levels of BNP are correlated well with the parameters of right heart catheterization (RHC); and thus, serve as an essential tool for risk stratification and disease management [5]. However, few studies have explored the clinical significance of BNP in patients with PH secondary to chronic lung disease [6,7,8]. Moreover, elevated BNP levels have been reported to predict mortality among patients with chronic lung diseases [6,7,9].

Among lung transplant candidates, possible associations of elevated BNP levels with high mean pulmonary artery pressure (MPAP) and mortality have not been investigated. We hypothesized that determination of BNP in waitlisted lung transplant candidates can serve as a useful tool for lung allocation and decision making regarding the necessity of RHC. Thus, we evaluated associations of N-terminal pro-B-type natriuretic peptide (NT-proBNP) levels with demographic and clinical characteristics, the parameters of the lung function test (LFT) and RHC, and mortality in patients on a lung transplant waiting list.

## 2. Materials and Methods

### 2.1. Study Population and Design

The study was designed as a retrospective investigation of patients treated at Rabin Medical Center, the national center for lung transplantation in Israel since 1997. The study was carried out in accordance with the Declaration of Helsinki and was approved by the institutional Ethics Committee. Figure 1 illustrates the study design. We identified 226 patients who were placed on a lung transplantation waiting list during the period November 2017–December 2019. After exclusion of 17 patients with class 1 PH, 2 patients with Eisenmenger’s syndrome, and 2 patients without available NT-proBNP measurement, 205 patients were included in the cohort. According to international guidelines for lung transplantation [10], waitlisting for transplant required preserved left ventricular ejection fraction and no evidence of significant structural left heart disorder on echocardiographic examination. For analysis of demographic, clinical, LFT, RHC, and all-cause mortality data, the patients were stratified into the following three groups: (1) patients who underwent transplantation; (2) survivors waiting transplantation; (3) patients who died waiting for transplantation.

### 2.2. Follow-Up Protocol

Lung transplant candidates were routinely chosen according to the International Society for Heart and Lung Transplantation (ISHLT) guidelines [10]. Plasma NT-proBNP determination, LFT, and RHC were performed on each patient before placement on the transplantation waiting list. The following parameters of LFT were obtained: forced expiratory volume in one second (FEV1), forced vital capacity (FVC), FEV1/FVC ratio, total lung capacity (TLC), residual volume, and diffusing lung capacity for carbon monoxide (DLCO). The following data of RHC were collected: MPAP, pulmonary capillary wedge pressure (PCWP), cardiac index (CI), cardiac output (CO), and pulmonary vascular resistance (PVR). The lung transplant candidates were evaluated in the ambulatory clinic every 3–4 months, on average. More frequent visits were scheduled according to clinical need. Follow-up included a detailed medical interview, physical examination, and LFT. Computed tomography of the chest and echocardiographic examinations were performed routinely every 12 months. The study outcome was all-cause mortality following placement on the transplantation waiting list. Vital status was registered, based on information from electronic medical records and the registry of the Ministry of Internal Affairs. Follow-up ended on 1 April 2021.

### 2.3. NT-proBNP Laboratory Analysis

NT-proBNP venous blood samples were collected into tubes containing separating gel and analyzed within eight hours. The NT-proBNP concentrations were quantified using the validated immunoassay (Elecsys^®^ proBNP Cobas e 602 analyzer, Roche Diagnostics, Indianapolis, IN, USA) [11]. The test principle was a sandwich immunoassay with total duration of 18 min. On first incubation with antigen in the sample, a biotinylated monoclonal NT-proBNP-specific antibody and a monoclonal NT-proBNP-specific antibody labeled with a ruthenium complex form a sandwich complex. On second incubation, after the addition of streptavidin-coated microparticles, the complex becomes bound to the solid phase via interaction of biotin and streptavidin. The reaction mixture is aspirated into the measuring cell where the microparticles are magnetically captured onto the surface of the electrode. Application of a voltage to the electrode then induces chemiluminescent emission, which is measured by a photomultiplier. According to the literature, an optimal strategy for diagnosing and assessing the prognosis of acute HF comprises three cut-off points: 450, 900, and 1800 pg/mL, for ages < 50, 50–75, and >75 years, respectively. This method yielded 90% sensitivity and 84% specificity [11].

### 2.4. Statistical Analysis

The results were expressed as means and standard deviations for quantitative data, and as numbers (percentages) of presented patients for qualitative data. We used the chi-square test to compare categorical variables and ANOVA to compare continuous variables. Statistical comparisons were performed between the data obtained for the study groups. Pearson’s correlation coefficient (*r*) was calculated to evaluate correlations of NT-proBNP levels with age, six-minute walk test distance, and parameters of LFT and RHC. *p*-values of ≤0.05 were considered significant. The area under the curve of the receiver operating characteristic plot was calculated to determine the relation of NT-proBNP with MPAP. Risk factors for death in waitlisted lung transplant candidates were evaluated by competing risk regression analysis using the Fine and Gray model. The effects of the covariates: age, gender, body mass index [BMI], and serum creatinine, on the association of NT-proBNP with case-specific mortality were evaluated by the Cox proportional hazards model with competing risk regression. We also created survival curves for waitlisted lung transplant candidates, using NT-proBNP levels, according to the Fine and Gray model. In a corresponding regression analysis, lung transplantation was considered a competing risk to mortality. Statistical analysis was performed using SAS software version 9.2 (SAS Institute Inc., Cary, NC, USA).

## 3. Results

### 3.1. Baseline Characteristics

Table 1 compares baseline characteristics of 205 lung transplant candidates, grouped according to receipt of transplantation (*n* = 97), and survival (*n* = 66) or death (*n* = 42) waiting for the transplantation. In the entire sample, the mean age was 59.3 ± 10.4 years; 66.3% were males. The most frequent reasons for lung transplantation were interstitial lung disease and chronic obstructive pulmonary disease (COPD). Non-survivors waiting for transplantation were more likely to have interstitial lung disease, while survivors who waited for transplantation more often presented with COPD. The majority of patients who underwent transplantation were males. Among non-survivors who waited for transplantations, the mean six-minute walk test distance, and the mean values of TLC and DLCO were significantly lower compared to patients in the other groups.

Patients with ILD had significantly lower FVC and TLC while patients with COPD had significantly lower FEV1 (Table 2).

Table 3 depicts correlations between baseline NT-proBNP levels and other parameters. Positive correlations of NT-proBNP concentrations were observed with age (*r* = 0.16, *p* = 0.02), FVC (*r* = 0.16, *p* = 0.02), and MPAP (*r* = 0.40, *p* < 0.001). The NT-proBNP values negatively correlated with DLCO (*r* = −0.24, *p* < 0.001) and CI (*r* = −0.29, *p* < 0.001). No significant correlations were found of NT-proBNP with the six-minute walk test distance, FEV1, TLC, and PCWP.

Figure 2 demonstrates the relation of NT-proBNP with MPAP. The optimal cut-off of NT-proBNP for prediction of MPAP levels > 35 mmHg was 251 pg/mL; with 58.1% sensitivity, 85.7% specificity, 45.0% positive predictive value, and 91.0% negative predictive value.

Analyzing the relation of NT-proBNP with CI, according to ≥2.5 and <2.5 L/min/m^2^, yielded a lower area under the curve, 0.628.

### 3.2. Survival Analysis

#### 3.2.1. Univariate Analysis

During the follow-up period, which extended up to 3.9 years (median 2.2 years), 70 of 205 (34.1%) waitlisted lung transplant candidates died. On univariate analysis, the following variables were significantly associated with increased survival: male sex (HR 0.42, 95% CI 0.23–0.75, *p* = 0.003), higher DLCO (HR 0.94, 96% CI 0.93–0.98, *p* = 0.001), and a longer six-minute walk test distance (HR 0.99, 95% CI 0.99–0.99, *p* = 0.07). While older age (HR 1.03, 95% CI 1.01–1.06, *p* = 0.04), higher BMI (HR 1.06, 95% CI 1.01–1.12, *p* = 0.04), and higher NT-proBNP (HR 1.38, 95% CI 1.08–1.78, *p* = 0.01) were significantly associated with decreased survival. The most common causes of death were infection and exacerbation of underlying lung disease.

Figure 3 illustrates survival curves for waitlisted lung transplant candidates stratified by NT-proBNP levels, according to the Fine and Gray model. Among patients with NT-proBNP levels > 250 compared to ≤250 pg/mL, 1 year (51.2% vs. 39.0%), 2 year (74.4% vs. 68.3%), and 3 year (90.7% vs. 82.9%) mortality rates were significantly higher (*p* = 0.004).

#### 3.2.2. Multivariate Analysis

On re-evaluation of NT-proBNP, gender, age, creatinine level and BMI by multivariate logistic regression, higher NT-proBNP (HR 1.49, 95% CI 1.10–2.03, *p* = 0.01) and male sex (HR 0.42, 95% CI 0.23–0.75, *p* = 0.003) were the variables most significantly associated with decreased and increased survival, respectively.

## 4. Discussion

In the present study, we investigated the prognostic significance of NT-proBNP levels among waitlisted lung transplant candidates. The main novelty of our investigation is the demonstration of a strong association of elevated NT-proBNP values with severe PH and with an increased risk of mortality. Since 2005, the Lung Allocation Score (LAS) has been implemented as the donor allocation system in the United States [12]. This score is a composite measure of several parameters including recipient age, underlying diagnosis, comorbidities, LFT parameters, and right-sided heart pressures. The LAS was based on national registry data and designed to minimize waiting list mortality, while maximizing post-transplant survival, thereby ensuring both patient and graft longevity [12]. BNP is not entered into the LAS score and its prognostic significance has not been studied among lung transplant candidates. To the best of our knowledge, this is the first report in the medical literature to demonstrate that NT-proBNP can serve as a useful biomarker for improving risk stratification among waitlisted lung transplant candidates. Indeed, we found that higher NT-proBNP levels were strongly associated with an increased risk of cumulative mortality. Our finding of an association between higher BNP values and increased mortality concurs with studies on patients with interstitial lung disease [6], COPD [9], and other various chronic lung diseases [7]. Notably, COPD was one of the frequent reasons for lung transplantation among our patients. In a meta-analysis of 2788 patients with COPD, elevated NT-proBNP levels predicted increased risk of all-cause mortality, regardless of the presence of HF [9]. In the present study, the patients had overall poor prognosis, which was related to the severity of the underlying lung disease. In this context of reduced life expectancy, NT-proBNP identified the patients with the highest mortality risk, regardless of the severity of the impairment of lung function. We suggest that including proBNP assessment in the LAS score may improve risk stratification for lung candidate prioritization and should be evaluated in further studies.

The underlying mechanisms for BNP elevation in chronic lung diseases are PH and right ventricular dysfunction [6,7,13]. In our institute, in accordance with international guidelines, patients with significant left heart disease are not registered for lung transplantation. Thus, in the study population, increased levels of NT-proBNP probably reflect the overload on the right ventricle, secondary to underlying pulmonary disease.

An additional interesting observation in the present study is the positive correlation between NT-proBNP and MPAP measured by RHC. We found that NT-proBNP levels below 251 pg/mL yielded 91.0% negative predictive value for MPAP above 35 mmHg. However, the relation between NT-proBNP and MPAP should be interpretated cautiously due to the relatively small sample size. Other studies of patients with chronic lung disease showed correlations of serum BNP with indices of right ventricular dysfunction on echocardiographic examination, Ref [6] and with a value of MPAP above 35 mmHg on RHC [7]. However, the gold standard for measuring MPAP is RHC [13,14]. Identifying patients with high MPAP (> 35 mmHg) before lung transplantation is essential because these patients will hemodynamically benefit from bilateral lung transplantation. The main disadvantage of RHC is its invasiveness and associated morbidity (1.1%) and mortality (0.05%) [14]. Furthermore, in many lung-transplantation centers, the LAS is not used, and RHC rather than used routinely, is performed selectively, in patients with clinical suspicion for high MPAP.

Currently, the main non-invasive method for estimating MPAP in chronic lung diseases is transthoracic echocardiography (TTE) [13]. Two systematic reviews reported low accuracy for assessment of PH by TTE in patients with chronic lung disease [15,16]. On the one hand, TTE tended to overestimate PH, leading to unnecessary RHC [15]. On the other hand, an opposite pattern was reported, by which 40% of patients with interstitial lung disease were misclassified as having a low probability of PH on TTE, when PH was confirmed at a subsequent RHC [17]. The decreased accuracy in assessing PH by TTE in chronic lung disease is explained by anatomical changes that affect imaging quality and the parameters measured by TTE [18,19]. These changes include a marked increase in intrathoracic gas, consolidation of lung tissue, expansion of the thoracic cage, and alternation of the position of the heart [18,19]. Moreover, in large transplantation centers, TTE is usually performed by different cardiologists and using non-uniform equipment.

Summarizing our findings and the data from the medical literature, we suggest that determination of NT-proBNP can serve as a useful tool in conjunction with TTE, for decision making regarding performing RHC in lung transplant candidates. Thus, in patients with low probability of significant PH (low NT-proBNP levels, without evidence of clinical signs of right HF and right ventricular dysfunction on TTE), RHC might be unnecessary.

The current study has several strengths. First, the heterogeneous cohort reflects a real-world population of waitlisted lung transplant candidates. Second, all the patients underwent RHC. Third, the prognostic significance of NT-proBNP was evaluated, taking into account the crucial parameters that may affect levels of NT-proBNP, such as age, gender, BMI, and serum creatinine. The limitations of this study include its retrospective, single-center design, which focused on only a single determination of NT-proBNP at the time of placement on the transplantation waiting list. The timing of patients’ registration for transplantation was according to international guidelines for ambulatory care. However, a few patients were registered urgently during hospitalizations in which they had acute exacerbations of their chronic lung disease. The exit from a steady state might have influenced the NT-proBNP level. Prospective larger studies are needed to validate our results, and to assess NT-proBNP in a serial manner until transplantation.

## 5. Conclusions

NT-proBNP determination may improve risk stratification among waitlisted lung transplant candidates. A higher value of NT-proBNP indicates an increased risk of mortality, and thus requires earlier allocation for lung transplantation. We suggest integrating this parameter in the LAS score. In addition, NT-proBNP can be useful in assessing the need for RHC in lung transplant candidates.

## Figures and Tables

**Figure 1 diagnostics-12-02112-f001:**
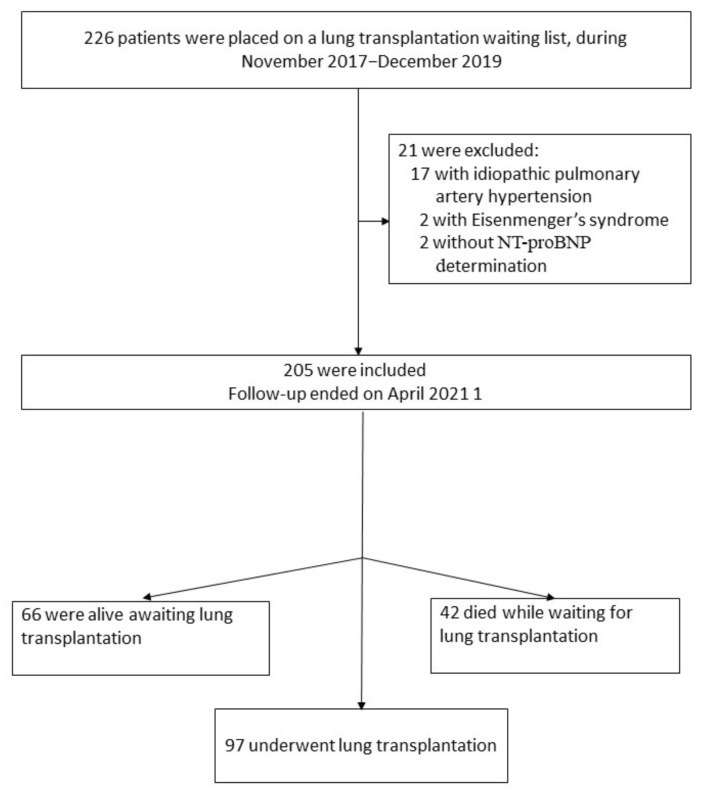
Flowchart presenting the study design.

**Figure 2 diagnostics-12-02112-f002:**
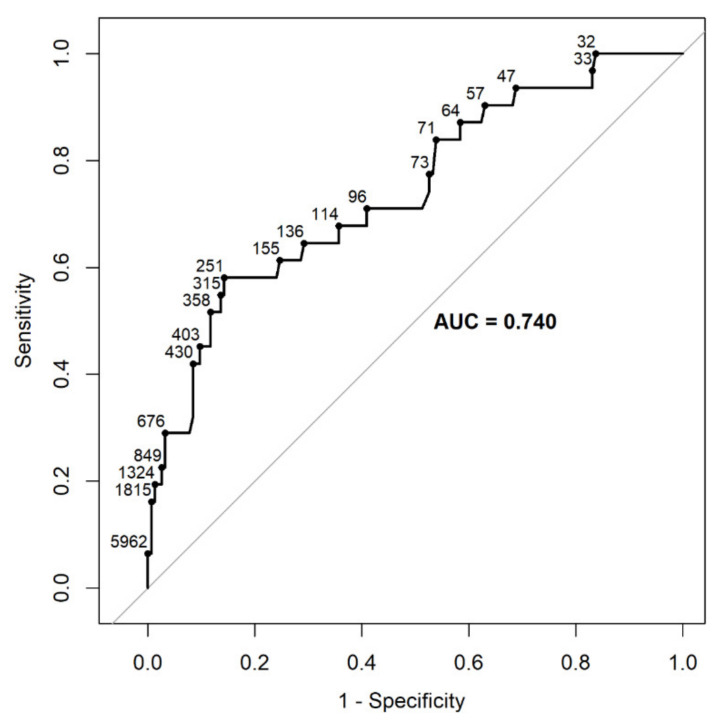
The relation of NT-proBNP levels and values of mean pulmonary artery pressure measured on right heart catheterization The receiver operator characteristic plot demonstrates sensitivity, specificity, and the area under the curve for the relation of NT-proBNP levels with values of mean pulmonary artery pressure measured on right heart catheterization ≤ 35 or >35 mm Hg.

**Figure 3 diagnostics-12-02112-f003:**
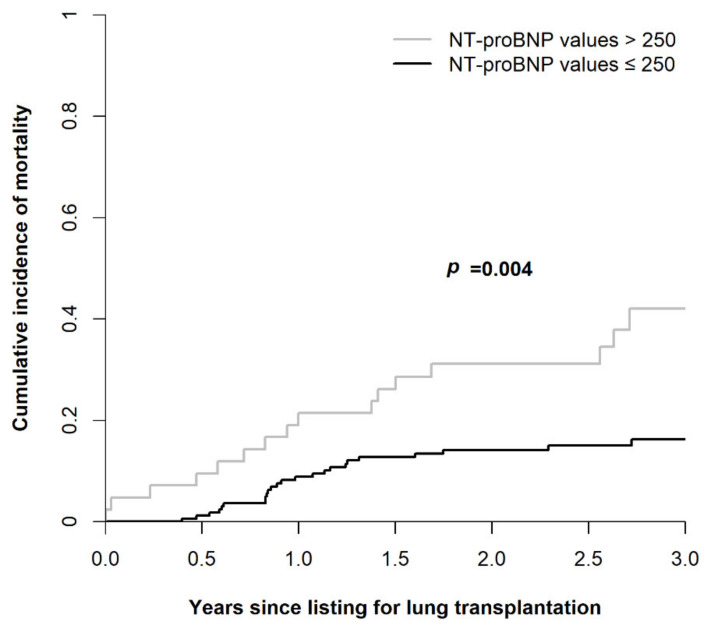
Cumulative incidence of mortality since listing for lung transplantation.

**Table 1 diagnostics-12-02112-t001:** Baseline characteristics of waitlisted lung transplant candidates stratified according to receipt of transplantation, and survival or death while waiting for the transplantation.

Ch Patients’ Characteristics	Total Population Results (*n* = 205)	Survivors Waiting Transplantation (*n* = 66)	Non-Survivors Waiting Transplantation (*n* = 42)	Lung Transplant Recipients (*n* = 97)	*p*-Value
Age (years)	59.3 ± 10.4	61.6 ± 8.7	62.0 ± 7.1	56.6 ± 11.9	<0.001
Male sex	136 (66.3%)	41 (62.1%)	20 (47.6%)	75 (77.3%)	0.002
Reason for transplantation					0.001
Interstitial lung disease	111 (54.1%)	27 (41.0%)	29 (69.0%)	55(56.7%)	
Idiopathic pulmonary fibrosis	62	15	16	31	
Autoimmune ILD	10	3	1	6	
Hypersnsitivity pneumonitis	11	2	3	6	
I-NSIP	11	4	5	2	
Occupational ILD	9	1	1	7	
Drug induced ILD	3	1	1	1	
Sarcoidosis	3	1	1	1	
Others	2	0	1	1	
COPD	72 (35.1%)	35 (53.0%)	12 (28.6%)	25 (25.8%)	
Non cystic fibrosis bronchiectasis	9 (4.4%)	2 (3.0%)	0 (0.0%)	7 (7.2%)	
Cystic Fibrosis	6 (2.9%)	1 (1.5%)	0 (0.0%)	5 (5.2%)	
Graft versus host disease	5 (2.4%)	1 (1.5%)	0 (0.0%)	4 (4.1%)	
Others	2 (1.0%)	0 (0.0%)	1 (2.4%)	1 (1.0%)	
Body mass index (kg/m^2^)	26.9 ± 5.6	26.1 ± 6.4	28.5 ± 5.6	26.6 ± 5.0	0.09
NT-proBNP (pg/mL)	287.6 ± 911.8	142 ± 224	571 ± 1698	263 ± 671	0.054
Serum creatinine (normal 0.5–0.9 mg/dL)	0.8 ± 0.5	0.8 ± 0.3	0.9 ± 0.7	0.9 ± 0.5	0.46
Six-minute walk test distance (meters)	283 ± 121	287 ± 127	227 ± 132	304 ± 106	0.006
Data of lung function test					
FEV1 (% of predicted value)	39 ± 18	37± 19	39 ± 18	40 ± 18	0.52
FVC (% of predicted value)	49 ± 16	52 ± 17	45 ± 14	48 ± 16	0.09
FEV1/FVC ratio	0.7 ± 0.3	0.6 ± 0.3	0.7 ± 0.3	0.7 ± 0.2	0.21
TLC (% of predicted value)	79 ± 37	89 ± 37	70 ± 36	76 ± 36	0.01
RV (% of predicted value)	137 ± 97	158 ± 93	119 ± 97	131 ± 99	0.09
DLCO (% of predicted value)	33 ± 12	35 ± 13	28 ± 10	35 ± 13	0.01
Data of right heart catheterization					
MPAP (mmHg)	25.1 ± 9.9	23.2 ± 8.8	25.0 ± 9.3	26.5 ± 11.0	0.15
PCWP (mmHg)	10.6 ± 6.7	9.6 ± 6.4	10.3 ± 7.7	11.4 ± 6.4	0.26
CI (L/min/m^2^)	2.4 ± 0.6	2.5 ± 0.6	2.4 ± 0.6	2.4 ± 0.6	0.32
CO (L/min)	4.5 ± 1.3	4.5 ± 1.2	4.2 ± 1.2	4.6 ± 1.4	0.21
PVR (WU)	3.2 ± 0.1	3.2 ± 2.0	3.9 ± 2.3	4.0 ± 4.2	0.37

Data are presented as means ± standard deviations or numbers (percentages) of presented patients. Abbreviations: ILD, interstitial lung disease; I-NSIP, idiopatic non sprcific interstitial pneumonia; COPD, chronic obstructive pulmonary disease; NT-proBNP, N-terminal pro-B-type natriuretic peptide; FEV1, forced expiratory volume in one second; FVC, forced vital capacity; TLC, total lung capacity; RV, residual volume; DLCO, diffusing lung capacity for carbon monoxide; MPAP, mean pulmonary artery pressure; PCWP, pulmonary capillary wedge pressure; CI, cardiac index; CO, cardiac output; PVR, pulmonary vascular resistance; WU, Wood units. Bold entries in the table idicate a *p*-value of ≤ 0.05.

**Table 2 diagnostics-12-02112-t002:** Baseline lung function tests of all lung transplant candidates, and separately for those with ILD and COPD.

Data of Lung Function Test	Total Population of the Study (*n* = 205)	Patients with ILD (*n* = 101)	Patients with COPD (*n* = 72)	*p*-Value
FEV1 (% of predicted value)	39 ± 18	51 ± 16	26 ± 9	<0.0001
FVC (% of predicted value)	48	47 ± 14	54 ± 15	<0.0001
FEV1/FVC ratio	0.67 ± 0.2	0.87 ± 0.1	0.39 ± 0.1	<0.0001
TLC (% of predicted value)	79 ± 37	51 ± 15	117 ± 21	<0.0001
RV (% of predicted value)	137 ± 97	63 ± 31	228 ± 66	<0.0001
DLCO (% of predicted value)	33 ± 12	33 ± 12	31 ± 12	0.07

Data are presented as means ± standard deviations or numbers (percentages) of presente patients. Abbreviations: ILD, interstitial lung disease; COPD, chronic obstructive pulmonary disease; FEV1, forced expiratory volume in one second; FVC, forced vital capacity; TLC, total lung capacity; RV, residual volume; DLCO, diffusing lung capacity for carbon monoxide.

**Table 3 diagnostics-12-02112-t003:** Correlations between levels of NT-proBNP and other baseline parameters.

Parameter	*r*	*p*-Value
Age	0.16	0.02
Six-minute walk test distance (meters)	−0.14	0.06
Data of lung function test		
FEV1 (% of predicted value)	0.12	0.1
FVC (% of predicted value)	0.16	0.02
TLC (% of predicted value)	0.01	0.9
DLCO (% of predicted value)	−0.26	<0.001
Data of right heart catheterization		
MPAP (mmHg)	0.4	<0.001
PCWP (mmHg)	0.12	0.11
CI (L/min/m^2^)	−0.29	<0.001

Abbreviations: NT-proBNP, N-terminal pro-B-type natriuretic peptide; FEV1, forced expiratory volume in one second; FVC, forced vital capacity; TLC, total lung capacity; DLCO, diffusing lung capacity for carbon monoxide; MPAP, mean pulmonary artery pressure; PCWP, pulmonary capillary wedge pressure; CI, cardiac index. Bold entries in the table idicate a *p*-value of ≤0.05.

## Data Availability

The data supporting the findings of this study are available from the corresponding author upon reasonable request.

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
