# Peer review of "Prognostic Significance of the N-Terminal Pro-B-Type Natriuretic Peptide in Lung Transplant Candidates on the Waiting List"

_diagnostics, 2022, doi:10.3390/diagnostics12092112_

Round 1

Reviewer 1 Report

I am very interested in the study entitled “Prognostic Significance of the N-terminal pro-B-type natriuretic peptide in Lung Transplant Candidates on the waiting list.” By Shimon Izhankian. I raised several points outlined below.

Patient characteristics is important, I think authors should show the detailed data as to the reason for lung transplantation. Detailed classification of interstitial lung disease should be shown ; J Clin Med. 2022 Mar; 11(6): 1747. Etiologic Classification of Diffuse Parenchymal (Interstitial) Lung Diseases.

Among lung transplantation recipients, 25 patients were classified as others, I think this is too large, and authors should show the detailed information as to the "others". 

Authors showed as emphysema, but COPD maybe used, I think.

I think pulmonary function test should be shown as to the  all of patiensts, interstitial lung disease only, and COPD only. Because the pattern of PFT with COPD and ILD is different.

I think authors should show the %DLOC/%VC and %DLCO/VA for pulmonary function test.

I think authors should show the data to affect the levels of NT-proBNP, for example, hypertension and CKD. 

In "3.2 suivival analysis", I think authors should analyze using the multivariate analysis. and if possible authors should show the survival curve relating to the level of NT-proBNP. I think this part is very important in this paper, however, the data lacks to show the prognostic significance of NT-proBNP. 

I think, if possible, authors should show the cause of death.

Although authors show the relationship with NT-pro BNP and PH, authors did not show the data as to left-side cardiac function. I think authors should show the left-side cardiac function using the echocardiagrapfic parameters.

In the discussion, authors wrote as “In our cohort, patients with significant left heart disease were excluded from the analysis, ..”, but in the materials and methods, there is no comment as to the exclusion of left heart disease.

In the discussion, authors wrote as “Three crucial decisions are made based on MPAP determination. 200 First, measurement of MPAP helps to choose single or bilateral lung transplantation. Second, MPAP evaluation is useful for planning extracorporeal lung support. Third, MPAP 202 is one of the LAS parameters.”, is this the thought of authors, or is there any data or paper to support the idea ? 

Author Response

We attaced a response to reviewer letter

Reviewer 2 Report

I read with interest the article by Izhakian et Col, titled "Prognostic Significance of the N-terminal pro-B-type natriuretic peptide in Lung Transplant Candidates on the waiting list".

The article is a monocentric retrospective series adressing the prognostic impact of a single blood NTproBNP measurment at listing in 205 lung transplant recipient (LTR), mostly ILD and COPD patients. The time on the waiting is long and mortality on the list very high (n=42). The autor obatined logical results correlating the results of NT por BNP levels with diverse  Pulmonary function and catheterism parameters, which were systematically performed. They also establihed ROC curve for the prediction of mPAP at listing and reported that nTproBNP<250 (very low levels) could exclude mPAP>35 mmHg (significantly elevated). AT last, on multivariate analysis NTproBNP levels correlated with mortality on the list. The article is easy to read, well written. Methods are clear and tables are clear (minor comment for Table 1: an additional colomn with the total population results would be nice and a gap of one line was inadvertantly inserted in the first 10 lines of the table: to be corrected). Discussion is well written.

I have some comments  in order to take the best for further decision making in LTR, from those data:

- in the introduction, some comments about the cinetic of NT proBNP would be nice. Indeed, this is a single assessment of NT porBNP, which can vary very quickly from one day to another and has sometimes been used as a follow up biomarker of overload. Morever in the methods, the autors could precise how many time after an exacerbation they waited before to sample NTBNP, i.e: are we at steady state (which is not easy in LTR)? Notably, Standard Deviation of the data is elevated suggesting a high variability between patients and this deserve further work in order to be sure that the result is meaning ful (for example, patients sampled during an acute exacerbation should be elminated, because the bad prognosis could not be related to the BNP, but related to the exacerbation frequency ....) 

- As noted by the autors on line 230, the main weakness of this work, to me, is the single assesment, especially in a population woth a very long time on the list. The autor se the LT listed candidates in the clinic every 3 mont: did they perform serial assesment of NTBNP? If yes, this would be a valuable inforamtion to add those data in a time dependent Cox Model. If not, the paragraph on limitations should be developped and this could be suggested to do in further works.

- In the discussion, another important missing point is : how the autors deal with this information for their LT candidates. Is the LAS used in Israel? is there a high emergency LT program? Indeed, accumulating negativ prognostic factors is a few help if it does not change the allocaiton system, considering the, again, very high mortality on the list.

- Figure 2: This part of the work is not the most important. It should be commented in the discussion that AUC of 0.74 is a not that discriminant, so the result is not that strong, especially in a 200 pts population. The interpretation in the discussion should be extremely cautious, depite I totally agree with the comments line 219-223: in our team catheterisation is far not systematical Morever another analysis (inversely) correleting NTproBNP with Cardiac index would also be interesting (perhaps more than mPAP?).

Author Response

We attached response to reviewer letter

Reviewer 3 Report

In general, the manuscript is well witten regarding the scientific aspects and format.

My comments:

1. The authors state that there is a positive correlation between FVC and nt-proBNP. This means the higher the FVC is, the higher the nt-proBNP.

Is there a mistake here?  If there is no mistake, how do the authors explaing the positive correlation of FVC and nt-proBNP?

2. If the specificity of nt-proBNP is high (85%), then its PPV is expected to be high too. However, it is stated to be 45%.

Likewise, if sensitivity is low (58%), then the NPV is expected to be low too but it is found to be 91%.

Specificity= TN/TN+FP  .......    PPV= TP/TP+FP

Sensitivity: TP/TP+FN    .......    NPV= TN/TN+FN

The authors sould correct the mistake or should explain the above findings in the Discussion.

3. Please check Table 1 carefully. The lines seem to be moved upward and downwards.

Author Response

We attached respone to reviewer letter

Round 2

Reviewer 3 Report

The manuscript has been fully revised according to my comments.